# Access Denied? The Unintended Consequences of Pending Drug Pricing Rules

Alan Kaplan [1],* , David J. Stewart [2], Gerald Batist [3], Silvana Spadafora [4], Sandeep Sehdev [2] and Shaun G. Goodman [5]

1  Enhanced Care Clinic, Aurora, ON L4G 1N2, Canada
2  Division of Medical Oncology, The Ottawa Hospital, University of Ottawa, Ottawa, ON K1Y 4E9, Canada; dstewart@toh.ca (D.J.S.); sehdev@mac.com (S.S.)
3  Segal Cancer Centre, Jewish General Hospital, Montreal, QC H3T 1E2, Canada; gerald.batist@mcgill.ca
4  Algoma District Cancer Program, Sault Ste. Marie, ON P6B 0A8, Canada; spadaforas@sah.on.ca
5  St. Michael's Hospital, Department of Medicine, University of Toronto, Toronto, ON M5S 1A1, Canada; goodmans@chrc.net
*  Correspondence: for4kids@gmail.com; Tel.: +1-905-883-1100

**Abstract:** The government of Canada now plans to bring into force new federal drug pricing regulations on 1 July 2022. We do not take issue with the goal of medication affordability, which is vital in healthcare the world over. Our concern is that the new guidelines are being implemented without due consideration for three major unintended consequences: regulatory changes will lower the number of clinical trials for new medications in Canada, fewer clinical trials will mean lower research and development investments, and changes will reduce patients' access to new medications. Access to effective medications is a cornerstone of healthcare for Canadian patients. As physicians, our duty to patient care demands that we tell the government to protect the right of Canadians to timely access to life-changing medicines.

**Keywords:** oncology; pharmacotherapy; funding; research; development; outcomes; PMPRB; access; clinical trials





## 1. Introduction

The government of Canada announced in 2019 that it would change the rules by which the Patented Medicine Prices Review Board (PMPRB) sets a price cap on medicines in Canada. The PMPRB is a quasi-judicial body created in 1987 out of concern that a stronger patent protection for medicines might cause the prices of these medicines to rise unacceptably and become unaffordable to Canadians. As such, the PMPRB has a regulatory mandate to prevent pharmaceutical patentees from charging consumers excessive prices during the patentees' statutory monopoly period [1].

Through the proposed guidelines, the federal government seeks to change Canada's drug pricing regime in three ways: to introduce economic tests to establish prices, to replace countries that have higher drug prices with countries that have lower prices in the PMPRB's international price comparisons, and to require drug manufacturers to report confidential information on rebates negotiated with government and private insurers [2].

The federal government contends that the proposed guidelines are intended "to provide transparency and predictability to patentees regarding the triage and review process" that the PMPRB uses to assess whether a patented medicine appears to be priced excessively in Canada [1]. As a result of the changes, the prices for new and innovative drugs in Canada are expected to drop significantly. At least, this is the assumed intended consequence of the proposed guidelines, especially in light of the proposed move to replace countries that have higher drug prices with countries that have lower prices in the PMPRB's international price comparisons.

Even this consequence, however, is open to debate. The Gastrointestinal Society, in its December 2020 Impact Report, contended that the long-term cost of innovative and generic medications may not decrease as a result of the changes; paradoxically, these long-term costs could in fact increase because of a contraction in supply. Put simply, the report's authors contended that the PMPRB's proposed changes "could lead to generic manufacturers stopping their supply to Canada, as the government sets their drug prices as a percentage of the patented drugs. If drastic cuts apply to patented drugs, then the cascade to generic drugs could leave Canadians without generic medicines as well." [3].

As physicians, we do not take issue with the goal of medication affordability, which is vital in healthcare the world over. However, we are deeply concerned about the unintended consequences of the federal government's planned new regulations [4]. More precisely, our concern is that the new guidelines are being implemented without due consideration for three major unintended consequences (Table 1). Those of us who would like the most up-to-date and effective medications to support our oncology patients should be aware of these consequences.

**Table 1.** The federal government's planned changes to drug pricing rules come with hidden costs.

| Three Major Unintended Consequences of Pending Drug Pricing Rules |
| --- |
| 1. Fewer clinical trials for new medicines, especially for cancer therapies, and therefore fewer opportunities for patients to gain early access to breakthrough treatments. |
| 2. Decrease in investment by pharmaceutical and life science companies in research and development. |
| 3. Reduction in access to innovative new medicines by doctors and their patients. |

## 2. Pending Regulatory Changes Will Lower the Number of Clinical Trials for New Medications in Canada

A 2019 study published in the European Journal of Health Economics showed that pharmaceutical manufacturers delay the introduction of new products in countries with highly regulated prices [5]. Life Sciences Ontario surveyed pharmaceutical leaders in 2019 and found that virtually every executive expects the proposed regulatory changes to result in "no launch" decisions and delayed medication launches in Canada [6].

At a time when 36 percent of current clinical trials are for cancer therapies [7], these leaders expect oncology to be among the treatment areas most affected by the changes. Companies that do not prioritize marketing a drug in a particular country are also less likely to approach that country for participation in clinical trials. Conducting trials in a country is a mechanism to familiarize local clinicians with the agent and to smooth the way for eventual launch.

The government's proposed changes are already having an impact. Since the initial announcement of the regulatory changes, the number of new trials registered with Health Canada has declined by more than 52 percent compared to the average in previous years [8].

## 3. Fewer Clinical Trials Will Mean Lower Research and Development Investments

Canada prides itself as a home to scientific innovation. The proposed changes regulating the pricing of pharmaceuticals risk undermining not only Canada's reputation as a world leader in life science research, but also our health-research infrastructure. If manufacturers view Canada's economic restrictions as unsustainable to market new treatments, pharmaceutical companies will take their research investments and product trials elsewhere.

## 4. Pending Changes Will Reduce Patients' Access to New Medications

Clinical trials are important because they test the efficacy of treatments, help determine appropriate dosages, and uncover any side effects. No less vital, participation in clinical trials is a way by which patients can access promising new agents that would otherwise be

unavailable to them. A reduction in clinical trials means impaired access to new drugs, and who among us would not want the most up-to-date and effective medications to support our oncology patients?

Access to effective medications is a cornerstone of healthcare for Canadian patients. Yet, compared to Americans, Canadians already face long delays before they can access effective new medications. For example, from 2008 to 2013, companies did not apply to Health Canada for drug approval until an average of 10 months after applying to the US and 8 months after applying to Europe [9]. From 2012 to 2019, new drugs were approved in Canada an average of 1.3 years after approval in the US and Europe [10]. After drug approval by Health Canada, it takes on average more than one year until most provincial health plans fund a medication [11]. The proposed regulatory changes will substantially worsen these delays.

Physicians and patients are not alone in concerns about drug-access impediments. The Canadian Society for Intestinal Research commissioned an impact report in which it stated, among other concerns, "We foresee significant delays for Canadians to access important new therapies that are lifesaving or life-altering" [4].

### 5. Concerned Organizations Are Speaking Up

The good news is that voices such as ours are having an impact. The new guidelines were set to take effect in 2020, but they were first delayed until 1 January 2021, then again until 1 July 2021, and again until 1 January 2022 [12]. In December 2021, the federal government announced that the coming into force of the guidelines would be postponed yet again—this time to 1 July 2022 [13]. In addition, as a result of more than 60 formal submissions that underscored problems with the guidelines, the PMPRB decided in mid-December not to proceed with some aspects of the reforms [14].

In its most recent announcement, the government pointed to two reasons for the latest postponement. The first was the persistence of the COVID-19 pandemic and the emergence of new coronavirus variants of concern. The government indicated that stakeholders must be allowed additional time to continue their focus on the pandemic recovery. The second reason was the "ongoing initiatives and considerations within an evolving pharmaceutical landscape." According to the government, this evolution is being propelled by the recent launch of Canada's Biomanufacturing and Life Sciences Strategy, progression of the National Strategy for Drugs for Rare Diseases, and development of a Canadian Drug Agency [15].

### 6. Physicians Must Be Heard

We agree with the government on its decision and the need for a further postponement. This time of change and uncertainty calls for further engagement and greater consensus; it calls for everyone with a stake in these changes to recognize that the pandemic has exposed the fragility of our healthcare system, and it calls for stakeholders to understand that the unintended consequences of the new guidelines will exacerbate this fragility, not remedy it.

In the coming weeks, we will continue to emphasize that the federal government's new regulatory changes are burdened with formidable unintended consequences: opportunities for patients to participate in clinical trials will be reduced, significant research and development investment in Canada will be lost, and access to medications will be reduced substantially [16]. Indeed, after two years of patient-treatment delays due to COVID-19, after Canada scrambled to acquire PPE and life-saving vaccines because our country lacked the domestic manufacturing capacity, the federal government should be working to make Canada more appealing to scientific innovation, not less.

As physicians, our duty to patient care demands that we remind the federal government of its responsibility to protect the right of Canadians to timely access to life-changing medicines. Not only must we avoid the encumbrances to patient access by opposing the PMPRB plans as proposed, but we must also encourage the emerging plans for major government investments in rebuilding post-pandemic healthcare systems and in health

science innovation, for our patients' sake. The time has arrived to tell the government to halt the planned implementation of these changes and reconsider them with more input from the physician and patient communities. Inform your colleagues about these proposed changes and their implications for patient care. Contact your member of parliament. Speak to your local and national news organizations. Educate your patients about what is at stake. Make your voice heard.

**Funding:** This research received no external funding.

**Acknowledgments:** The authors would like to thank and acknowledge the writing assistance provided by the group Voice for Access.

**Conflicts of Interest:** A.K. has no conflicts of interest related to this work but has been on speaker bureaus or advisory boards for Astra Zeneca, Behring, Boehringer Ingelheim, Cipla, Covis, Eisai, GSK, Merck Frosst, NovoNordisk, Novartis, Sanofi Genzyme, Teva, Trudel, and Valeo. D.J.S. has no conflicts directly related to this work but does have the following disclosures in that he has been involved in advisory boards or consultations for AbbVie Canada, Astra Zeneca, Merck, Roche, and Canadian Agency for Drug and Technologies in Health. In addition, the Ottawa Hospital receives research support from a broad range of pharmaceutical companies. He does have a US Patent no. 9.675.663 (test to predict response to TUSC2/FUSI gene therapy). G.B. has no conflicts of interest related to this work but collaborates with many academic, biotech, and pharmaceutical enterprises in clinical trials and new treatment developments. He derives no personal stake nor remuneration from them. There are no conflicts of interest in relation to this publication. S.S. (Sandeep Sehdev) has no conflicts of interest related to this work but has been on advisory boards or speaker bureaus for Astra Zeneca, Lilly, Merck, Novartis, Pfizer, Roche, and SeaGen. S.S. (Silvana Spadafora) has no stated conflicts of interest. S.G.G. has no conflicts of interest related to this work but has research grant support (e.g., steering committee or data and safety monitoring committee) and/or speaker/consulting honoraria (e.g., advisory boards) from Amgen, Anthos Therapeutics, AstraZeneca, Bayer, Boehringer Ingelheim, Bristol Myers Squibb, CSL Behring, Daiichi-Sankyo/American Regent, Eli Lilly, Esperion, Ferring Pharmaceuticals, HLS Therapeutics, JAMP Pharma, Merck, Novartis, Novo Nordisk A/C, Pendopharm/Pharmascience, Pfizer, Regeneron, Sanofi, Servier, and Valeo Pharma and has salary support/honoraria from the Heart and Stroke Foundation of Ontario/University of Toronto (Polo) Chair, Canadian Heart Research Centre and MD Primer, Canadian VIGOUR Centre, Cleveland Clinic Coordinating Centre for Clinical Research, Duke Clinical Research Institute, New York University Clinical Coordinating Centre, PERFUSE Research Institute, and TIMI Study Group (Brigham Health).

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
