# Peer review of "Access Denied? The Unintended Consequences of Pending Drug Pricing Rules"

_curroncol, doi:10.3390/curroncol29040204_

Round 1

Reviewer 1 Report

I think it’s a well written manuscript, with reputable authors who have knowledge about the details and implications of change of rules of PMPRB. Their expressed opinion also seem to be shared by other organizations (e.g. Gastrointestinal Society). I believe that the content is appropriate for Current Oncology, as PMPRB changes can result in significant setbacks to the capacity of oncology clinical trials in Canada, and deprive many Canadians access to new cancer drugs.

Some specific feedback

  • Though it’s clear that clinical trial capacity, R&D, and patient/physician access to new innovative medications can be reduced from PMPRB changes, I hope the authors can elaborate on the why PMPRB wanted to make changes in the first place, to provide readers with a context, and to offer a more balanced view. For example, it was mentioned that “the price of new innovative drugs in Canada are expected to drop significantly [with the proposed changes]”, but there was no subsequent mention of this point. Perhaps authors can comment on what alternatives measures can be done, if PMPRB proposed changes were to be blocked, to decrease the price of new drugs. There are some excellent points made in other similar articles (https://badgut.org/wp-content/uploads/PMPRB-Impact-Report-2021.pdf). They suggest that the cost of medications would actually not decrease with the changes, but paradoxically increase, as the cost of generics will also increase (as generic drug cost is determined as a fixed % of cost of the original patented drug), and generic companies will also leave Canada.
  • Perhaps authors can consider re-organize their 3 points about major unintended consequences of the pending drug pricing rules. Specifically, I believe that their point 1 and 3 are really the same point. To some extent, point 2 is also related to point 1 and 3.
  • Perhaps one point that was not described is the relevance of PMPRB in the 21st century. It appears that other agencies (such as CADTH) have already taken on more rigorous roles than PMPRB in determining drug prices

Author Response

Dear Reviewer,

Thank you very much for your time, insights and kind words. We, the authors of the article, are deeply grateful for your careful attention and your suggestions.

We have taken action to address one of your three suggestions:

Yes, we agree with your suggestion to point out that the proposed changes may indeed lead to an increase in long-term costs for patented and generic medications. We also appreciate the reference you supplied and we will refer to it directly.

With respect to your two other suggestions:

  • We have chosen not to change the three-point structure of the article. This change would necessitate a significant rewrite and we simply don't have the time and resources to carry it out. We do, however, appreciate the suggestion and the kind words on how well written the article is.
  • We have chosen not to address the relevance of the PMPRB in the 21st This point is thoughtful, but we believe it is not germane to this article.

We hope this note explains our position clearly.

Thank you for your excitement and enthusiasm about the article. And, again, thank you very much for your time and thoughtful suggestions.

Yours sincerely,

Alan  Kaplan MD CCFP(EM) FCFP CPC(HC)

First author, signing  for all of the authors

Reviewer 2 Report

This is a well written opinion piece which is rather important for Canadian oncologists to be aware of the ramifications of the change in legislation coming for new drug approval. The authors have set out a realistic view of what the proposed changes could affect future drug approval. My only comment would be, it would be nice to know how other physicians could provide a voice to this. I am sure there will be readers who would want to know how they could go about lobbying for adjustments to these changes. Otherwise, it was good read which I enjoyed.

Author Response

Dear Reviewer,

Thank you very much for your time, insights and kind words. We, the authors of the article, are deeply grateful for your careful attention and your suggestion.

With respect to your suggestion, we have chosen not to add more text to the end of the article to instruct other physicians to go about lobbying for adjustments to these changes. We feel we have exhausted the call to action in the current draft. To wit, the call to action urges physicians to inform their colleagues, express their concerns to their members of Parliament, speak with local and national media, and educate their patients.

We hope this note explains our position clearly.

Thank you for your excitement and enthusiasm about the article. And, again, thank you very much for your time and thoughtful suggestions.

Yours sincerely,

Alan  Kaplan MD CCFP(EM) FCFP CPC(HC)

First author, signing  for all of the authors